# Shikonin Inhibits Cell Growth of Sunitinib-Resistant Renal Cell Carcinoma by Activating the Necrosome Complex and Inhibiting the AKT/mTOR Signaling Pathway

**DOI:** 10.3390/cancers14051114

**Published:** 2022-02-22

**Authors:** Sascha D. Markowitsch, Olesya Vakhrusheva, Patricia Schupp, Yasminn Akele, Jovana Kitanovic, Kimberly S. Slade, Thomas Efferth, Anita Thomas, Igor Tsaur, René Mager, Axel Haferkamp, Eva Juengel

**Affiliations:** 1Department of Urology and Pediatric Urology, University Medical Center Mainz, Johannes Gutenberg University Mainz, 55131 Mainz, Germany; sascha.markowitsch@unimedizin-mainz.de (S.D.M.); olesya.vakhrusheva@unimedizin-mainz.de (O.V.); p.schupp@imb-mainz.de (P.S.); yakele@students.uni-mainz.de (Y.A.); jkitanov@students.uni-mainz.de (J.K.); kimberlysue.slade@unimedizin-mainz.de (K.S.S.); anita.thomas@unimedizin-mainz.de (A.T.); igor.tsaur@unimedizin-mainz.de (I.T.); rene.mager@unimedizin-mainz.de (R.M.); axel.haferkamp@unimedizin-mainz.de (A.H.); 2Institute of Pharmaceutical and Biomedical Sciences, Johannes Gutenberg University Mainz, 55128 Mainz, Germany; efferth@uni-mainz.de

**Keywords:** renal cell carcinoma (RCC), sunitinib, therapy resistance, shikonin (SHI), growth, necroptosis, AKT

## Abstract

**Simple Summary:**

Advanced renal cell carcinoma (RCC) remains an incurable disease, despite the establishment of new therapeutic options during the last decades. Unfortunately, drug resistance inevitably evolves, making better treatment strategies essential. Shikonin (SHI) from traditional Chinese medicine has shown promising antitumor properties with other tumor entities. Thus, in the current investigation, we evaluated the impact of SHI on therapy-sensitive and therapy-resistant RCC cells. SHI led to significant inhibition of progressive tumor cell growth in both therapy-sensitive and therapy-resistant RCC cells. This was accompanied by cell cycle arrest and reduction in cell cycle activating proteins, which results in blockage of cell division. SHI also activated necrosome complex proteins, leading to necroptosis, a programmed cell death. Furthermore, SHI inhibited the AKT/mTOR pathway, pivotal for cell survival and growth. Thus, SHI may hold promise in improving the treatment of RCC, even in its advanced form.

**Abstract:**

Therapy resistance remains a major challenge in treating advanced renal cell carcinoma (RCC), making more effective treatment strategies crucial. Shikonin (SHI) from traditional Chinese medicine has exhibited antitumor properties in several tumor entities. We, therefore, currently investigated SHI’s impact on progressive growth and metastatic behavior in therapy-sensitive (parental) and therapy-resistant Caki-1, 786-O, KTCTL-26, and A498 RCC cells. Tumor cell growth, proliferation, clonogenic capacity, cell cycle phase distribution, induction of cell death (apoptosis and necroptosis), and the expression and activity of regulating and signaling proteins were evaluated. Moreover, the adhesion and chemotactic activity of the RCC cells after exposure to SHI were investigated. SHI significantly inhibited the growth, proliferation, and clone formation in parental and sunitinib-resistant RCC cells by G2/M phase arrest through down-regulation of cell cycle activating proteins. Furthermore, SHI induced apoptosis and necroptosis by activating necrosome complex proteins. Concomitantly, SHI impaired the AKT/mTOR pathway. Adhesion and motility were cell line specifically affected by SHI. Thus, SHI may hold promise as an additive option in treating patients with advanced and therapy-resistant RCC.

## 1. Introduction

Renal cell carcinoma (RCC) is the most common cancer of the kidney, accounting for 80–90% of kidney cancer, and is the most aggressive urologic cancer [1,2,3]. The incidence of RCC has increased worldwide in the last two decades [1,4], with the highest incidence occurring in the western world [5]. In 2020 the number of new RCC cases was estimated to have exceeded 400,000 worldwide, with a mortality of ~180,000 [6]. RCC is often initially asymptomatic, resulting in late diagnosis, and approximately 25–30% of patients have distant metastases at initial diagnosis [7]. Another 15–30% of patients develop metastases during therapy. Cure, once metastasis has occurred, is extremely rare. Current therapeutic approaches are based on molecular targets, specifically tyrosine kinase inhibitors (TKIs) and immune checkpoint inhibitors (ICIs). Sunitinib is a TKI that has contributed to significant improvement in the treatment of patients with advanced RCC, particularly in first-line therapy [8]. Unfortunately, this treatment leads only to disease stabilization in terms of prolongation of progression-free and overall survival [7]. Despite the benefits of targeted therapies for patients with metastatic (advanced) RCC, the long-term prognosis remains poor, with a 5-year survival rate of 17% [9]. Resistance to established therapeutics is mainly responsible for this poor outcome [10,11].

The limited gain in lifetime with approved treatment options has led to desperation in many patients. Consequently, most tumor patients desire the inclusion of naturopathic procedures into their therapeutic regimen or are already using them [12,13,14]. Especially substances from traditional Chinese medicine (TCM), used for thousands of years, are gaining increased interest. This need for the use of natural substances is only satisfied to a limited extent in everyday clinical practice since scientific evidence for the efficacy of the substances is often lacking, and thus their use is not justifiable [12,13,14].

Shikonin (SHI), extracted from the dried root of *Lithospermum erythrorhizon*, exhibits antitumor properties in various tumor entities [15,16,17]. It has been shown to enhance other antitumor therapeutics [18] and resensitize therapy-resistant lung [19,20], gastric [21], bladder [22], prostate [23], and leukemia cells [24] to therapy. The exact mechanism of enhancement and/or resensitization is still largely unknown. Preliminary evidence shows that SHI acts on signaling pathways that are jointly responsible for cell cycle regulation, resulting in cell cycle arrest [25]. Moreover, we and others have shown that SHI induces various forms of regulated cell death, such as apoptosis and necroptosis, in different cancer cells [23,26,27,28]. Necroptosis is a caspase-independent form of programmed cell death, regulated by the necrosome complex proteins RIP1, RIP3, and MLKL. An inhibitory effect of SHI on the AKT signaling pathway, key to (cancer) cell survival, has also been demonstrated [19,29,30]. We have shown in previous investigations that the expression and activity of AKT increases with the emergence of therapy resistance in RCC in vitro and in vivo [31,32,33].

Therapy resistance in RCC cells is associated with increased invasive behavior [34], unrestrained tumor cell growth [31,32,33,35], and further dedifferentiation [35]. Evidence has been provided that combining targeted therapeutics with natural compounds not only delays but abrogates therapy resistance [23,31,35,36,37,38]. The effects of two of these natural substances, sulforaphane (from broccoli) [37,38,39,40] and artesunate (from annual mugwort) [36,41], have been investigated. Since data reflecting the impact of SHI on RCC is scant and nonexistent for therapy-resistant RCC, the present study was designed to investigate whether SHI might have potential in treating therapy-sensitive (parental) and, more importantly, RCC that has become resistant to sunitinib.

## 2. Materials and Methods

### 2.1. Cell Cultures

Sunitinib-sensitive and sunitinib-resistant Caki-1, 786-O, KTCTL-26, and A-498 cell lines were obtained and cultured as previously described [41]. Umbilical cords were provided by the Department of Gynecology at the University Medical Center Mainz, 55131 Mainz, Germany. Human umbilical vein endothelial cells (HUVEC) were extracted from the umbilical vein, and cultured in M199 medium (Sigma-Aldrich, Darmstadt, Germany) supplemented with 20% FCS (Gibco, Thermo Fisher Scientific, Darmstadt, Germany), 0.8% ECGS (endothelial cell growth supplement) (PromoCell, Heidelberg, Germany), 5000 IU heparin (Merck, Darmstadt, Germany), 50 mg gentamycin (Biozym, Hessisch Oldendorf, Germany), 1% Anti/Anti (Gibco, Thermo Fisher Scientific, Darmstadt, Germany), and 20 mm HEPES-buffer (Sigma-Aldrich, Darmstadt, Germany). Subcultures from passages 1 to 5 were used.

### 2.2. Resistance Induction and Application of Sunitinib and Shikonin

Sunitinib resistance was induced by gradually increasing the exposure of cells to sunitinib (0.1–1 µM) (free Base, Massachusetts LC Laboratories, Woburn, MA, USA) over a period averaging 10 weeks. Once resistant, cells were continuously exposed to 1 µM sunitinib. For details, see the previously published description of resistance induction [41]. Shikonin (SHI) (Sigma-Aldrich, Darmstadt, Germany), dissolved in DMSO (10 mM stock solution), was diluted with the cell culture medium to concentrations ranging from 0.5 to 2.5 μM. Controls remained untreated. The IC50 (concentration of inhibitor that reduces the response by half) of SHI in parental and sunitinib-resistant subcells was evaluated using 72 h growth data. Possible toxic effects of the compounds were evaluated by trypan blue (Sigma-Aldrich, Darmstadt, Germany).

### 2.3. RCC Cell Growth and Proliferation

Cell growth and proliferation were determined as previously described [41]. In brief, cell growth of parental and sunitinib-resistant cells, treated for 24, 48, and 72 h with SHI (0.5–2.5 µM), was analyzed with an MTT assay. Dose-response kinetics normalized to the mean cell number after 24 h incubation are shown in percent.

Cell proliferation was determined by colorimetric quantification of incorporated BrdU using the BrdU Cell Proliferation Assay (Calbiochem/Merck Biosciences, Darmstadt, Germany). Values were normalized to the untreated controls and shown as a percentage.

### 2.4. Clonogenic Growth Behavior

The clonogenic assay was performed as previously described [41]. Briefly, parental and sunitinib-resistant RCC cells were treated with SHI (0.1–0.5 µM). Grown colonies were visualized with Coomassie Blue G250 (Sigma-Aldrich, Darmstadt, Germany) and quantified with a biomolecular imager (Sapphire, Azure Biosystems, Biozym, Hess. Oldendorf, Germany). Values were normalized to the untreated control (set to 100%) and shown in percentage.

### 2.5. Cell Cycle Phase Distribution

Cell cycle analysis was evaluated as previously reported [41]. Briefly, RCC cells were incubated with SHI for 48 h and grown to sub-confluency. DNA was labeled with propidium iodide (50 µg/mL) (Invitrogen, Thermo Fisher Scientific, Darmstadt, Germany) and quantified by flow cytometer (Fortessa X20, BD Biosciences, Heidelberg, Germany). The proportion of cells in the G0/G1, S, or G2/M phases was compared to untreated control cells and expressed as a percentage.

### 2.6. RCC Cell Death

A FITC-Annexin V Apoptosis Detection Kit was used to evaluate apoptosis and necrosis (BD Biosciences, Heidelberg, Germany). After exposing cells to 1.5 µM SHI for 48 h, they were then treated with Annexin V-FITC and/or propidium iodide for 15 min. Quantification of apoptotic and necrotic events was performed by flow cytometer (Fortessa X20, BD Biosciences, Heidelberg, Germany). Analysis of 1 × 10^4^ collected events per sample was performed using DIVA software (BD Biosciences, Heidelberg, Germany). FlowJo software was used for further analysis (BD Biosciences, Heidelberg, Germany).

Necroptotic effects and the impact of caspases on growth inhibition were analyzed using 3-(4,5-dimethylthiazol- 2-yl)-2,5-diphenyltetrazolium bromide (MTT) dye. For more details, see “Tumor Cell Growth” (Section 2.3). Tumor cells were treated either with SHI (0.5–2.5 µM) alone or combined with necrostatin-1 (80 µM) (Sigma-Aldrich, Darmstadt, Germany), a RIP1 kinase inhibitor, for 24 h or with zVAD (20 µM) (Selleckchem, München, Deutschland), a multi-caspase inhibitor, for 48 h.

Caspase activity was determined using the Caspase-Glo^®^ 3/7, 8, and 9 assays (Promega Corporation, Madison, WI, USA), according to the manufacturer’s protocol. A total of 1 × 10^4^ cells/well were seeded into a 96-well-plate and incubated with SHI (1 µM) or zVAD (20 µM) (Selleckchem, München, Deutschland) for 6 h. Luminescence was measured using a multi-mode microplate reader (Tecan, Spark 10 M, Tecan, Grödig, Austria).

### 2.7. Protein Expression and Activity—Western Blot Analysis

To explore the expression and activity of cell cycle regulating proteins, cell death regulating proteins, and AKT/mTOR signaling proteins, Western blot analysis was performed. A total of 50 µg of each tumor cell lysate was applied to polyacrylamide gel (7% to 12%) and separated (10 min at 80 V followed by approximately 1 h at 120 V) until the dye front reached the lower gel end. Subsequently, proteins were transferred to PVDF membranes (1 h at 100 V). Membranes were blocked with 10% non-fat dry milk for 1 h. Membranes were then incubated overnight with the following primary antibodies directed against the cell cycle regulating proteins: p21 (rabbit IgG, clone 12D1, dilution 1:1000, Cell Signaling, Frankfurt am Main, Germany), p27 (mouse IgG_1_, clone 57/Kip1, dilution 1:500, BD Biosciences, Heidelberg, Germany), cyclin A (mouse IgG_1_, clone 25, dilution 1:500, BD Biosciences, Heidelberg, Germany), cyclin B (mouse IgG_1_, clone 18, dilution 1:1000, BD Biosciences, Heidelberg, Germany), CDK1 (mouse IgG_1_, clone 2, dilution 1:2500, BD Biosciences, Heidelberg, Germany), pCDK1 (rabbit, clone 10A11, dilution 1:1000, Cell Signaling, Frankfurt am Main, Germany), CDK2 (mouse IgG_2a_, clone 55, dilution 1:2500, BD Biosciences, Heidelberg, Germany) and pCDK2Thr160 (rabbit, polyclonal antibody, dilution 1:1000, Cell Signaling, Frankfurt am Main, Germany). β-actin (clone AC-1, dilution 1:10,000, Sigma-Aldrich, Taufkirchen, Germany) served as the internal control for the cell cycle regulating proteins, and protein bands were normalized to β-actin.

To detect AKT/mTOR signaling proteins, the following primary antibodies were used: AKT (mouse IgG_1_, clone 55, dilution 1:1000, BD Biosciences, Heidelberg, Germany), pAKTS473 (rabbit, clone D9E, dilution 1:1000, Cell Signaling, Frankfurt am Main, Germany), mTOR (rabbit, clone 7C10, dilution 1:1000, Cell Signaling, Frankfurt am Main, Germany), pmTORSer2448 (rabbit, polyclonal antibody, dilution 1:1000, Cell Signaling, Frankfurt am Main, Germany). Proteins were normalized to a total protein control, quantified by Coomassie brilliant blue staining, as previously described [41].

To evaluate the expression and activity of necrosome complex proteins, the primary antibodies RIP1 (rabbit IgG, clone D94C12, dilution 1:1000), pRIP1S166 (rabbit IgG, clone D1L3S, dilution 1:1000), RIP3 (rabbit IgG, clone E1Z1D, dilution 1:1000), pRIP3S227 (rabbit IgG, clone D6W2T, dilution 1:1000), MLKL (rabbit IgG, clone D2I6N, dilution 1:1000), and pMLKLS358 (rabbit IgG, clone D6H3V, dilution 1:1000) were used (all Cell Signaling, Frankfurt am Main, Germany). Proteins were normalized to a total protein control, quantified by Coomassie brilliant blue staining, as previously described [41].

Secondary antibodies were HRP-conjugated rabbit-anti-mouse IgG or goat-anti-rabbit IgG (IgG, both: dilution 1:1000, Dako, Glosturp, Denmark). AKT/mTOR signaling proteins and cell death regulating proteins were normalized to total protein. Signal visualization and quantification were performed as previously reported [23]. The exposure time was adapted to the signal intensity (device-specific maximum, >65,000 = over-saturated). Only images with a maximum band intensity below 65,000 were evaluated. Ratios of protein intensity divided by either β-actin intensity or by whole protein intensity were calculated and expressed in percentage, compared to untreated controls, set to 100%.

### 2.8. AKT Blockade

Cell growth after AKT blockade was evaluated using 3-(4,5-dimethylthiazol- 2-yl)-2,5-diphenyltetrazolium bromide (MTT) dye. For more details, see “Tumor cell growth” (Section 2.3). Cell growth was assessed after 48 h treatment with SHI (0.5–2.5 µM) or AZD5363 (20 µM), an AKT inhibitor. Untreated cells served as control (set to 100%).

### 2.9. Adhesion to Extracellular Matrix Proteins and Vascular Endothelium

To investigate the interaction of control and 72 h SHI-treated cells with extracellular matrix proteins (ECM), 24-well tissue culture plates were pre-coated with collagen G (200 µg/mL; Merck, Darmstadt, Germany), fibronectin (10 µg/mL; Gibco™, Thermo Fisher Scientific, Darmstadt, Germany) or laminin (10 µg/mL; Corning GmbH, Kaiserslautern, Germany) overnight at 4 °C. Wells covered with PBS served as a background control for the unspecific binding. To minimize non-specific cell adhesion, plates were blocked 1 h before assay with 1% BSA.

To determine the interaction of tumor cells with the vascular endothelium, 1.25 × 10^5^ HUVECs were plated into 24-well-plates 16 h prior to an adhesion assay. For adhesion assays 1.25 × 10^5^ parental and sunitinib-resistant RCC cells, pre-stained with CellTracker Green CMFDA Dye (Invitrogen™, Thermo Fisher Scientific, Darmstadt, Germany) and pre-treated with SHI or diluent, were used per well and incubated for 1 h (adhesion to ECM) or 2 h (adhesion to HUVEC) at 37 °C in a humidified CO_2_ incubator. Non-attached tumor cells were washed off with PBS, containing Mg^2+^ and Ca^2+^, and fixed with 2% glutaraldehyde. To score adhesion, the mean fluorescent intensity of attached cells was determined using a Sapphire Imager (Azure Biosystems, Munich, Germany). Values were presented as a percentage compared to untreated controls, set to 100%.

### 2.10. Tumor Cell Motility

Chemotactic activity and cell migration were assessed using Falcon^®^ 24-well Companion plates, and Corning^®^ FluoroBlok Inserts with 8 µm pore size (both: Corning GmbH, Kaiserslautern, Germany). To determine cell motility toward a soluble chemotactic agent, an FCS gradient was employed. For chemotaxis, inserts were left unaltered. To score cell migration, inserts were pre-coated with 200 µg/mL of collagen G overnight at 4 °C.

Aliquots of 6 × 10^5^ parental and sunitinib-resistant RCC cells, pre-stained with CellTracker Green CMFDA Dye (Invitrogen™, Thermo Fisher Scientific, Darmstadt, Germany) and 72 h pre-treated with SHI or diluent, were seeded per insert. The companion plates were loaded with cell growth media containing 30% FCS and inserts with the cell suspension in media with 10% FCS. After 24 h incubation, the inserts were washed with PBS, containing Mg^2+^ and Ca^2+^, and fixed with 2% glutaraldehyde. To quantify the number of cells migrating through the insert membrane toward the 30% FCS stimulus, mean fluorescent intensity on the lower surface of the inserts was determined using a Sapphire Imager (Azure Biosystems, Munich, Germany). Values were presented as a percentage compared to untreated controls, set to 100%.

### 2.11. Statistical Analysis

All experiments were performed at least three times. The evaluation and generation of mean values, the associated standard deviation, and normalization in percent were performed by Microsoft Excel (Office Professional Plus 2016, Microsoft, Redmond, WA, USA). Statistical significance was calculated with GraphPad Prism 7.0 (GraphPad Software Inc., San Diego, CA, USA): two-sided *t*-Test (Western blot, apoptosis, cell cycle, adhesion, tumor cell motility), one-way ANOVA test (BrdU, clonogenic assay), and two-way ANOVA test (MTT). Correction for multiple comparisons was performed using the conservative Bonferroni method. Differences were considered statistically significant at a *p*-value ≤ 0.05.

## 3. Results

### 3.1. Shikonin Inhibited Progressive Growth Behavior of Parental and Sunitinib-Resistant RCC Cells

To evaluate SHI’s antitumor potential, parental and sunitinib-resistant Caki-1, 786-O, KTCTL-26, and A-498 cells were exposed to ascending concentrations of SHI, 0.5 to 2.5 µM. Growth was significantly restricted in a time- and dose-dependent manner for both parental (Figure 1a,d,g,j) and sunitinib-resistant (Figure 1b,e,h,k) cells. Similar findings were observed when proliferation was assessed after 48 h incubation with SHI (Figure 1c,f,i,l). Parental and sunitinib-resistant Caki-1 cells showed a higher sensitivity to SHI than the other RCC cells, with IC50 values of 0.58 µM and 0.82 µM, respectively. In 786-O and KTCTL-26 cells, the antitumor activity of SHI was comparable (IC50 of 1.1 µM in parental, 1.63 µM in sunitinib-resistant 786-O cells and 1.34 µM in parental, 1.22 µM in sunitinib-resistant KTCTL-26 cells). A-498 cells displayed the lowest response to SHI with IC50 values almost twice those of Caki-1 cells (parental: IC50 = 1.54 µM, sunitinib-resistant: IC50 = 1.85 µM).

The growth inhibitory effects of SHI on parental and sunitinib-resistant RCC cells became particularly evident with clonogenic growth. A total of 0.2 µM SHI already significantly reduced the colony-forming capacity of parental and sunitinib-resistant Caki-1, 786-O, and KTCTL-26 cells (Figure 2a–c). Here, the previously less sensitive A-498 cells showed significantly reduced clonogenic growth after exposure to SHI as well, starting at a concentration of 0.3 µM SHI (Figure 2d).

### 3.2. Shikonin-Induced Cell Cycle Arrest in the G2/M Phase

Simultaneously with SHI-induced reduction in cell growth, proliferation, and colony-forming capacity in the RCC cells, significant cell cycle arrest occurred (Figure 3). In parental Caki-1 cells, SHI evoked a G0/G1 phase arrest, with a concomitantly diminished percentage of cells in the S and G2/M phases, whereas a G2/M phase arrest was evident in sunitinib-resistant Caki-1 cells (Figure 3a). In response to SHI, parental and sunitinib-resistant 786-O and KTCTL-26 cells were also significantly arrested in the G2/M phase in parental cells by a simultaneous significant reduction in the G0/G1 phase (Figure 3b,c). Moreover, SHI induced G2/M cell cycle arrest in parental A-498 cells but did not significantly affect cell cycle distribution in their sunitinib-resistant counterparts (Figure 3d).

Since SHI significantly affected the cell cycle phases in both parental and sunitinib-resistant Caki-1 and 786-O cells, these cell lines were employed for in-depth investigation regarding cell cycle regulation at the protein level. The shift in the cell cycle phases after exposure to SHI was accompanied by the corresponding modulation of cell cycle regulating proteins. In parental and sunitinib-resistant Caki-1 cells, SHI caused significantly diminished expression of cyclin B, CDK1, and pCDK1, responsible for progression into the G2/M phase (Figure 4a,e–g). The expression of cyclin A, CDK2, p21, and p27 was not affected by SHI in Caki-1 cells (Figure 4a–d,h).

In 786-O cells, SHI significantly down-regulated the expression of cyclin A and CDK1 in both parental and sunitinib-resistant cells (Figure 5a,d,f). Cyclin B was down-regulated after SHI application in sunitinib-resistant 786-O cells (Figure 5a,e). Expression of p27 was down-regulated in parental 786-O cells, and CDK2, as well as pCDK2, was enhanced in sunitinib-resistant 786-O cells after exposure to SHI (Figure 5a,c,h,i).

Since the strong growth inhibitory effects of SHI on the RCC cells could have been due to apoptotic or necroptotic processes, whether and to what extent this may have occurred was investigated.

### 3.3. Shikonin-Induced Apoptosis

Treatment with 1.5 µM SHI for 48 h resulted in a significant increase in apoptotic cells in parental and sunitinib-resistant Caki-1, 786-O, and KTCTL-26 cells (Figure 6a–c). In contrast, neither parental nor sunitinib-resistant A-498 cells were affected by SHI in regard to apoptosis (Figure 6d).

### 3.4. Shikonin-Induced Necroptosis in Parental and Sunitinib-Resistant RCC Cells

Treatment with necrostatin-1, a necroptosis inhibitor, significantly abolished the growth inhibitory effect of SHI in parental and sunitinib-resistant Caki-1, 786-O, KTCTL-26, and A-498 cells (Figure 7a–h). Only at higher SHI concentrations necrostatin-1 could no longer completely counteract this effect (Figure 7a–h). Mono-treatment with necrostatin-1 led to significantly elevated cell growth (Figure 7a–h).

These apoptosis results suggest that, at least for Caki-1, 786-O, and KTCTL-26, caspase-dependent cell death may be responsible for the growth inhibition induced by SHI. On the other hand, growth inhibition by SHI could be due to caspase-independent necroptosis, as shown by employing the necroptosis inhibitor necrostatin-1. To investigate caspase dependency in SHI-induced cell death and growth inhibition, the caspase activity of Caki-1 and 786-O cells was blocked with zVAD, a multi-caspase inhibitor, and tumor cell growth was evaluated. A total of 24 h incubation of parental and sunitinib-resistant Caki-1 cells with SHI or combined treatment with SHI-zVAD resulted in a comparable dose-dependent growth inhibitory effect (Figure 8a,b), indicating a caspase-independent growth inhibition. An approximate 50% growth inhibition was reached after treatment with 1 µM SHI in parental Caki-1 cells (Figure 8a), while 1.5 µM SHI was needed to comparably inhibit the sunitinib-resistant Caki-1 cells (Figure 8b). Neither in parental nor in sunitinib-resistant Caki-1 cells did combined treatment with SHI and zVAD abrogate the SHI-induced growth inhibition (Figure 8a,b).

In parental 786-O cells, SHI-zVAD application had no effect on SHI-induced growth inhibition (Figure 8c), again indicating a caspase-independent growth inhibition by necroptosis induction. In contrast, SHI-zVAD in sunitinib-resistant 786-O cells significantly counteracted the growth inhibitory effect of SHI at a concentration range of 1 to 2 µM SHI (Figure 8d), indicating at least partial caspase-dependent growth inhibition through apoptosis.

To further investigate caspase dependency, the activity of single caspases was evaluated, whereby the activated caspases 8, 3/7, and 9 were considered as key activators for apoptosis [42]. Activated caspase 8 has also been shown to be a necroptosis inhibitor. In parental and sunitinib-resistant Caki-1 cells, treatment with SHI (1 µM) significantly inhibited caspase 8 activity after 6 h (Figure 9b). Caspase 3/7 and 9 activity was not altered (Figure 9a,c). If treated with zVAD, the positive control, the activity of caspase 3/7 and 9 was reduced by more than 50% (Figure 9a,c). Caspase 8 activity of Caki-1 cells was not significantly affected by zVAD (Figure 9b).

In parental and sunitinib-resistant 786-O cells, the treatment with SHI resulted in a similar significant inhibition of caspase 8 activity (Figure 9e). Caspase 9 activity in sunitinib-resistant 786-O cells was moderately but significantly inhibited by SHI. In parental 786-O cells, no effect was apparent on caspase 9 activity after treatment with SHI (Figure 9f). In both parental and sunitinib-resistant 786-O subcell lines, SHI had no significant effect on caspase 3/7 activity (Figure 9d). zVAD alone again induced significant inhibition of all caspase activity.

SHI primarily induced caspase-independent cell growth inhibition, as well as reduced caspase 8 activity, indicating necroptosis induction. To evaluate this in depth, the expression and activity of key proteins of the necrosome complex, responsible for necroptotic genesis and progression, were examined in Caki-1 cells.

In parental and sunitinib-resistant Caki-1 cells, SHI time-dependently altered expression and activity (phosphorylation) of the necrosome complex proteins RIP1, RIP3, and MLKL. After 6 h SHI exposure in parental and sunitinib-resistant cells, the activity of RIP1 (pRIP1) significantly increased (Figure 10a,c). This effect was no longer significant (parental cells), or the activity was even significantly inhibited (sunitinib-resistant cells) after 24 h SHI exposure. Concomitantly, RIP1 expression significantly decreased (Figure 10a,b). RIP3 expression significantly increased after 6 h incubation with SHI in both parental and sunitinib-resistant Caki-1 cells. As with RIP1, 24 h exposure to SHI had no longer any effect on RIP3 expression in parental and sunitinib-resistant Caki-1 cells (Figure 10a,d). Despite increased expression of RIP3, pRIP3 expression was initially inhibited in parental and sunitinib-resistant cells after 6 h incubation with SHI (Figure 10a,e). However, after 24 h of SHI exposure, parental Caki-1 cells showed significantly increased RIP3 (pRIP3) activity (Figure 10a,e). In sunitinib-resistant cells, RIP3 activity did not significantly increase compared to the control, but the reduction in RIP3 activity observed after 6 h was reversed after 24 h (Figure 10a,e). MLKL activity (pMLKL) significantly increased in parental and, to a weaker extent, in sunitinib-resistant Caki-1 cells after 6 h SHI exposure, similar to the RIP1 activity (pRIP1) (Figure 10a,c,g). After 24 h, the effect was again no longer detectable. MLKL expression behaved contrarily, significantly increasing after 24 h in parental and by a trend in sunitinib-resistant Caki-1 cells (Figure 10a,f), but after 6 h exposure to SHI was not altered (parental cells), or even significantly inhibited (sunitinib-resistant cells) (Figure 10a,f). SHI thus exhibited time-dependent effects on both the expression and activity of the necrosome complex proteins of parental and sunitinib-resistant Caki-1 cells, further corroborating necroptosis induction.

### 3.5. Shikonin Impaired AKT/mTOR Signaling

SHI has been shown to inhibit cell growth via the AKT/mTOR signaling pathway [15], which regulates cell survival and growth. In the present investigation, the influence of SHI on AKT and mTOR was representatively measured in the Caki-1 and 786-O subcell lines. In parental and sunitinib-resistant Caki-1 cells, SHI significantly reduced the expression of AKT and mTOR in a dose-dependent manner (Figure 11a,c,e). In parental Caki-1 cells, 0.5 µM SHI and in sunitinib-resistant cells 1 µM SHI caused significant down-regulation. Administration of 1 µM SHI also caused a significant inhibition of AKT activity (pAKT) in sunitinib-resistant Caki-1 cells that, in trend, was also apparent in parental Caki-1 cells (Figure 11a,d). Only in parental Caki-1 cells did SHI significantly decrease mTOR activity (pmTOR) (Figure 11a,f).

In parental and sunitinib-resistant 786-O cells, SHI caused a dose-dependent, significant inhibition of AKT (pAKT) and mTOR (pmTOR) activity (Figure 11b,h,j). The total protein expression of mTOR was significantly affected by 1 µM SHI in these cells (Figure 11b,i). However, neither in parental nor in sunitinib-resistant 786-O cells did SHI affect the total protein expression of AKT (Figure 11b,g).

To evaluate the functional relevance of AKT during SHI-induced growth inhibition, tumor cell growth was investigated after pharmacological blocking of AKT through AZD5363. Inhibition of AKT signaling resulted in a significant reduction in cell growth, comparable to the growth inhibition after SHI treatment. The observed cell type-specific response to SHI was also evident after AKT blockade. Thus, sunitinib-resistant Caki-1 cells showed higher sensitivity (~30% growth inhibition) to the AKT inhibitor compared to parental cells (~15% growth inhibition). AKT inhibition in parental Caki-1 cells resulted in cell growth inhibition, comparable to treatment with 0.5 µM SHI (Figure 12a), whereas the growth reduction in sunitinib-resistant Caki-1 cells was comparable to treatment with 1.0 µM SHI (Figure 12b). The 786-O cells demonstrated a higher sensitivity toward the AKT inhibitor (Figure 12c,d). Again, inhibition of AKT signaling resulted in significantly reduced cell growth of parental and sunitinib-resistant cells. The growth inhibition induced by the AKT inhibitor was higher than 50% and corresponded with the growth reduction observed after treatment with 0.5 to 1.0 µM SHI (Figure 12c,d).

### 3.6. Shikonin Cell Line Specifically Influenced Adhesion and Motility

One reason for the limited success of RCC therapy is frequent metastasis at initial diagnosis. To investigate this aspect, the effect of SHI on the metastatic potential of Caki-1 and 786-O cells was analyzed. SHI did not alter adhesion, migration, and chemotaxis of parental and sunitinib-resistant Caki-1 cells (Figure 13a,b,e,f). However, parental 786-O cells revealed a significant increase in general adhesion capacity (PBS) and attachment to vascular endothelium (HUVEC cells) after treatment with SHI (Figure 13c). Parental and sunitinib-resistant 786-O responded with increased motility to SHI application (Figure 13g,h).

## 4. Discussion

Renal cell carcinoma is one of the most aggressive urologic tumors, and in an advanced, metastatic stage, a cure is rare. Standard targeted therapies for advanced RCC initially result in a significant increase in survival but ultimately culminate in evolved resistance. Hence, therapy is only palliative [43]. In the current study, sunitinib, a first-line TKI [8,43], was used in our resistance model. Since the phytochemical compound SHI has been shown to induce antitumor effects in various tumor entities [44,45,46,47,48,49], we assessed its potential to enhance therapy before and after resistance was developed in four RCC cell lines.

The influence of SHI depended on the cell line that was being investigated. Continuous cell growth and metastasis are well-known characteristics of advanced RCC [50]. However, no effect of SHI on cell adhesion was detected in parental or sunitinib-resistant Caki-1 cells. Cell migration and chemotaxis were also not affected by SHI in Caki-1 subcells. In contrast, adhesion to vascular endothelium, migration, and chemotaxis of parental as well as sunitinib-resistant 786-O cells even increased after SHI treatment. This appears decidedly unbeneficial, but concomitant with enhanced cell adhesion and motility, the clonogenic capacity of single cells was significantly decreased. Thus, even if SHI induces a higher partial adhesion and motility in 786-O cells, those cells no longer form clone colonies, a basic requirement to settle in other organs and allow distant metastases to develop. The inhibitory effect of SHI on clonogenic growth was very strong in both parental and sunitinib-resistant 786-O and Caki-1 cells, already in the nanomolar range (200–300 nM). Similar effects of SHI have been shown in other tumor entities. Treatment with SHI of retinoblastoma cells [51] and colorectal carcinoma cells [46,52,53] has also resulted in decreased cell colony formation. SHI not only reduced clonogenic growth in colon cancer cells but also led to lower cell viability and a less motile cell type [52]. SHI’s impact on the adhesive and migratory potential of RCC cells thus seems marginal, but a strong inhibitory effect on clonogenic growth capacity was apparent. Whether this appreciably influences metastatic spread deserves further investigation.

The antitumor effect of SHI on RCC cells was particularly evident in its inhibitory effect on progressive growth behavior. Exposure to SHI time- and dose-dependently inhibited the growth of all four tested RCC cell lines, both parental and sunitinib-resistant subcell lines. The IC50 varied between 0.5 and 1.9 µM SHI, with Caki-1 cells being the most sensitive. Studies of other urologic tumor entities report a working concentration with antitumor effects at 0.4 µM in bladder cancer [22] and a range of 0.5–10 µM in prostate carcinoma over a 24 h period [54]. In suitable accordance with the presently presented results, parental as well as therapy-resistant prostate carcinoma cells responded with a significant growth inhibition to SHI administration at a concentration of ~0.5 µM [23]. Studies of non-urologic tumors, such as non-small cell lung carcinoma, demonstrated 100% inhibition of cell viability in vitro at 2 µM SHI [55].

Besides inhibiting growth, SHI significantly reduced proliferation of all four tested RCC cell lines, both parental and sunitinib-resistant subcell lines. Although the effect on proliferation was less pronounced than on cell growth, it followed the dose-dependent course apparent for growth inhibition. SHI administration has also resulted in a dose-dependent proliferation reduction in prostate carcinoma cells [23]. Consistent with this, dose-dependent inhibition of proliferation by SHI was evident in glioma cells [56] and in the MCF-7 breast cancer cell line [57].

Concomitant with the SHI-inhibition of growth, a G2/M phase arrest occurred in parental and sunitinib-resistant 786-O, KTCTL-26, parental A-498, and sunitinib-resistant Caki-1 cells. Likewise, SHI induced a significant increase in the G2/M phase in breast cancer [58] and in prostate carcinoma [59], including refractory DU145 prostate carcinoma cells [23]. In contrast, parental Caki-1 cells showed an increase in cells in the G0/G1 phase after SHI application. Accordingly, cell cycle arrest in the G0/G1 phase after SHI administration has also been described in gallbladder tumor cells, which was accompanied by intrinsic apoptotic effects [44]. The number of colon cancer cells in the G0/G1 phase significantly increased with ascending SHI concentration [46].

On the molecular level, SHI treatment significantly reduced the expression of the cell cycle activating proteins cyclin B, CDK1, and pCDK1 in parental and sunitinib-resistant Caki-1 cells. In parental and sunitinib-resistant 786-O cells, SHI administration diminished the expression of cyclin A and CDK1. Moreover, SHI decreased cyclin B expression in sunitinib-resistant 786-O cells. Cyclin B or cyclin A, in complex with CDK1, were responsible for G2/M phase progression [60,61]. Thus, down-regulation of those proteins leads to cell cycle arrest in the SHI-treated RCC cells. Accordingly, analysis of hepatocellular carcinoma cells revealed that negative regulation of cyclin B and CDK1 led to cell cycle arrest in the G2/M phase [62].

Furthermore, SHI significantly increased the number of apoptotic cells in three (Caki-1, 786-O, and KTCTL-26) of the four tested RCC cell lines. Several reports described that SHI induced apoptosis [15] in colon cancer cell lines [63], pancreatic cancer [64], and non-small cell lung bronchial carcinoma tumor cells [65]. Inhibition of proliferation in breast cancer cell lines has also been attributed to apoptosis induction [66], and in colorectal cancer cells, administration of SHI increased ROS levels, which in turn induced intrinsic apoptosis [46].

Since annexin V staining used in the present investigation reflected extensive apoptotic events and likewise captured necroptotic events [67], necroptosis may have played a decisive role in SHI’s influence. Indeed, the most notable effect that SHI had was in regard to necroptosis in all four RCC cell lines, both parental and sunitinib-resistant cells. Co-treatment with necrostatin-1, a necroptosis inhibitor, completely abolished SHI’s effect in the present investigation. Therefore, it may be assumed that the reduction in RCC cell growth and proliferation after SHI administration was predominantly caused by necroptosis induction. A study on RCC has shown that necroptosis proteins are basally expressed in tumor tissue [68]. The authors postulated that this might be associated with a higher sensitivity to necroptosis induction. This could explain the increase in the growth of cells treated exclusively with necrostatin-1. Several in vitro studies have demonstrated that SHI induces necroptosis. In a previously published study with therapy-sensitive RCC cells, the growth inhibitory effect of SHI was also significantly abrogated by necrostatin-1 [69]. In prostate carcinoma [23] and pancreatic cancer [64], the antitumor effect of SHI was also reverted by necrostatin-1, indicating necroptosis. Moreover, combined treatment of SHI with zVAD, a multi-caspase inhibitor, did not abolish SHI’s growth inhibition in the RCC cells. This indicates caspase-independency being responsible for cell death, i.e., necroptosis. SHI-induced growth inhibition in lung cancer cells could not be abrogated by zVAD either [55]. SHI has also been shown to activate cell death in glioma cells via the necroptosis pathway [70]. Here, growth inhibition through SHI was likewise not affected after combined treatment with zVAD. In refractory prostate carcinoma cells, a caspase-independent activity of SHI has also been described and substantiated with a lack of caspase activity inhibition through zVAD [23]. However, in sunitinib-resistant 786-O cells, we found a partial effect of zVAD, indicating that SHI, at least in this cell line, might induce both necroptosis and apoptosis.

Inhibited caspase 8 was considered to indicate necroptosis induction [71]. In parental and sunitinib-resistant Caki-1 and 786-O cells, treatment with SHI caused a significant reduction in caspase 8 activity, whereas the activity of caspases 3/7 and 9 remained unchanged, further indicating necroptosis initiation in these RCC cells. In contrast, caspases 3/7 and 9 were unchanged after SHI administration, compared with the untreated control, further supporting the previous data pointing to necroptosis. zVAD, used as a positive control, inhibited the caspase activity, except for that of caspase 8, in parental and sunitinib-resistant Caki-1 cells. Caspase 9 is a key molecule of intrinsic apoptosis [72], whereas caspase 8 in its active form initiates extrinsic apoptotic cell death. Both caspases, 8 and 9, cleaved caspase 3, leading to its activation and induction of regulated cell degradation. In its activated form, caspase 8 thereby inhibits the necroptosis pathway by cleaving necrosome complex proteins RIP1 and RIP3 [73]. Thus, unchanged or inhibited caspase 8 activity by SHI, as shown in the present work, argues for caspase-independent non-apoptotic cell death. Under inactivated caspase 8 conditions, RIP1 and RIP3 phosphorylate each other and form the necrosome complex. Xiang et al. [71] have described the activation of necroptosis in liver carcinoma cells via inactivated caspase 8 and preserved RIP1 and RIP3 from caspase 8-dependent cleavage, thereby inducing necroptosis. In suitable agreement with the present findings, SHI has been shown to abrogate the activation of caspases 3 and 8 in multiple myeloma [74]. Furthermore, cessation of RIP1 cleavage with subsequent necroptotic cell death was reported in this study.

In the present investigation, a causality between caspase 8 inhibition and a corresponding increase in the necrosome complex proteins RIP1, RIP3, and MLKL, was demonstrated. Thereby, SHI time-dependently enhanced both the expression and activity of necrosome complex proteins in parental and sunitinib-resistant Caki-1 cells. Elevated protein expression of RIP1 and RIP3 by SHI has also been demonstrated in other tumor entities. In osteosarcoma, an increase in the total expression of RIP1 and RIP3 after treatment with SHI has been described [75], with RIP3, in particular, being strongly increased. In glioma cells, SHI induced a concentration-dependent increase in protein expression of RIP1 and RIP3 [27]. Moreover, SHI increased MLKL expression, which was augmented in a time-dependent manner [27]. In prostate carcinoma cells, SHI treatment has resulted in altered protein expression, particularly with the two necrosome complex proteins, RIP1 and RIP3 [23]. In nasopharyngeal carcinoma, SHI treatment increased the expression of all three necrosome complex proteins [28], whereas pretreatment with necrostatin-1 abrogated SHI-induced effects. Thus, in suitable accordance with other studies, the current investigation confirmed caspase-independent necroptosis induction after SHI treatment by deactivating caspase 8 and elevating necrosome complex protein activity.

SHI induced not only cell cycle arrest and necroptosis but also affected the AKT/mTOR signaling pathway, which is a key regulator of cell survival, cell growth, proliferation, and cell migration [76]. This signaling pathway is activated in many malignancies, including RCC, and promotes tumor progression [77]. Notably, treatment with SHI resulted in a significant reduction in AKT and mTOR expression and activity in parental and, even more strongly, in sunitinib-resistant Caki-1 and 786-O cells. Accordingly, in cervical cancer cells, SHI diminished the expression of AKT and mTOR [78]. SHI’s impact on the AKT/mTOR signaling pathway has also been shown in lymphoma [79]. Combined SHI-doxorubicin treatment synergistically increased this effect. Consistent with the present investigation, in prostate cancer cells, SHI effectively inhibited phosphorylation levels, i.e., activated AKT and mTOR [59]. In bronchial carcinoma cells, treatment with SHI also reduced mTOR expression and phosphorylation [80]. Moreover, SHI affected the expression and activity of AKT in lung cancer cells [81].

The SHI-induced AKT deactivation in parental and sunitinib-resistant RCC cells was involved in the growth inhibition observed under SHI application, as shown by pharmacological AKT inhibition. Thereby, the functional blocking of AKT in parental and sunitinib-resistant Caki-1 and 786-O cells induced a comparable growth inhibition to that induced by SHI. Prostate cancer cells treated with an AKT inhibitor have also shown reduced cell growth [82], and colorectal cancer cell growth has been inhibited by combined treatment with cetuximab and the AKT inhibitor AZD5363 [83]. In recent studies, AKT inhibition has been shown to be an effective treatment modality in gastric cancer cells [84] and breast cancer [85]. It may, therefore, be assumed that AKT inhibition is part of SHI’s antitumor action in the tested RCC cells.

## 5. Conclusions

SHI impaired the growth behavior of parental and sunitinib-resistant RCC cells. The mechanism of action was predominantly based on cell cycle arrest and necroptosis as well as inhibition of the AKT/mTOR signaling pathway. The degree to which each of these mechanisms of action contributed to the antitumor effect of SHI on Caki-1, 786-O, KTCTL-26, and A498 RCC cells was not uniform. However, since the antitumor effect of SHI was apparent in all four cell lines, we postulate that SHI could be a promising addition to standard therapy for patients with advanced and therapy-resistant RCC.

## Figures and Tables

**Figure 1 cancers-14-01114-f001:**
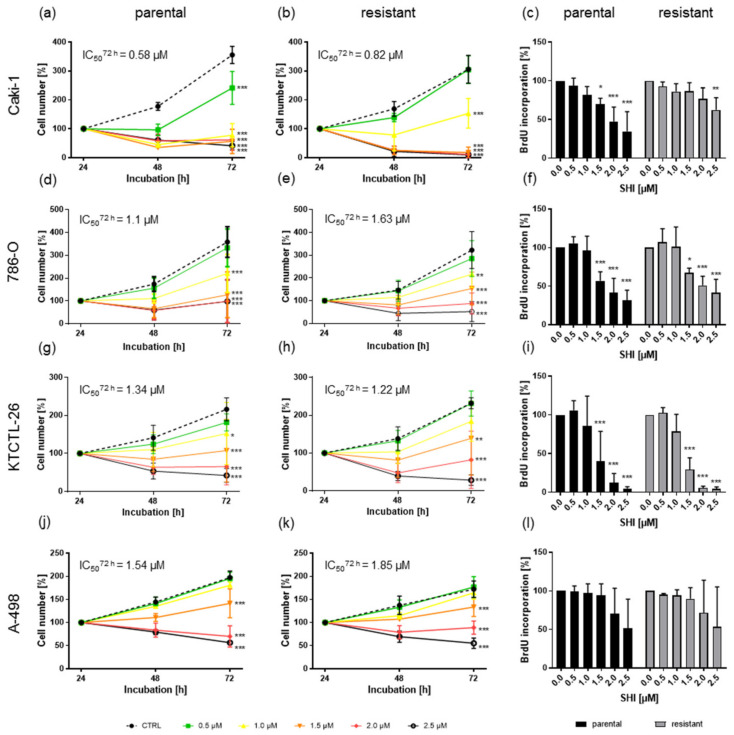
Growth of parental and sunitinib-resistant Caki-1 (**a**,**b**), 786-O (**d**,**e**), KTCTL-26 (**g**,**h**), and A-498 (**j**,**k**) cells after 24, 48, and 72 h exposure to SHI (0.5–2.5 µM). Cell number was set to 100% after 24 h incubation. Error bars indicate standard deviation (SD). Significant difference to untreated control: * *p* ≤ 0.05, ** *p* ≤ 0.01, *** *p* ≤ 0.001. *n* = 5. Proliferation of parental and sunitinib-resistant Caki-1 (**c**), 786-O (**f**), KTCTL-26 (**i**), and A-498 (**l**) after 48 h incubation with SHI (0.5–2.5 µM). Untreated controls were set to 100%. Error bars indicate standard deviation (SD). Significant difference to untreated controls: * *p* ≤ 0.05, ** *p* ≤ 0.01, *** *p* ≤ 0.001. *n* = 4.

**Figure 2 cancers-14-01114-f002:**
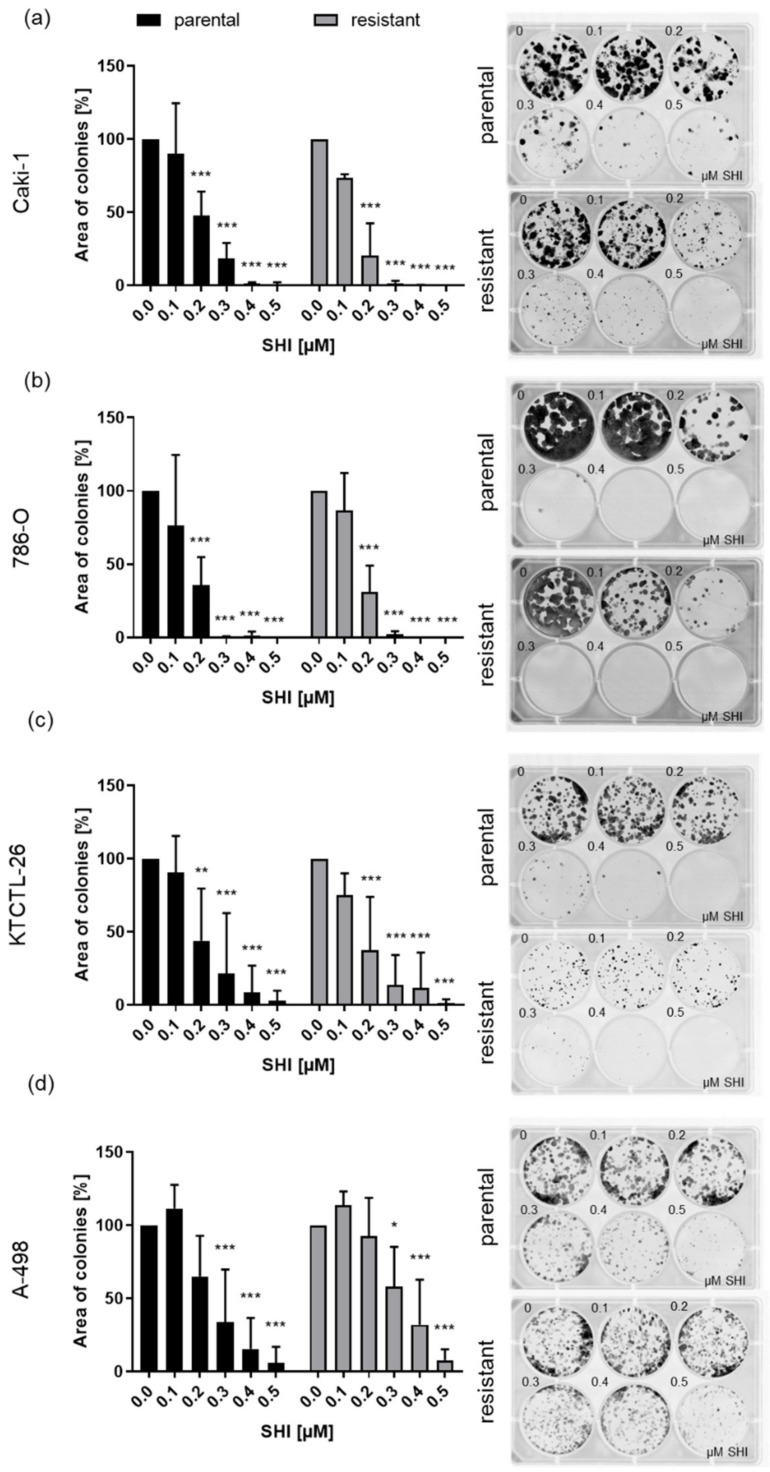
Clonogenic growth of parental and sunitinib-resistant Caki-1 (**a**), 786-O (**b**), KTCTL-26 (**c**), and A-498 (**d**) cells treated with SHI (0.1–0.5 µM) for 10 days. Untreated cells served as controls (set to 100%). Representative images of the clonogenic assays are shown on the right. Error bars indicate standard deviation (SD). Significant difference to untreated controls: * *p* ≤ 0.01, ** *p* ≤ 0.01, *** *p* ≤ 0.001. *n* = 5.

**Figure 3 cancers-14-01114-f003:**
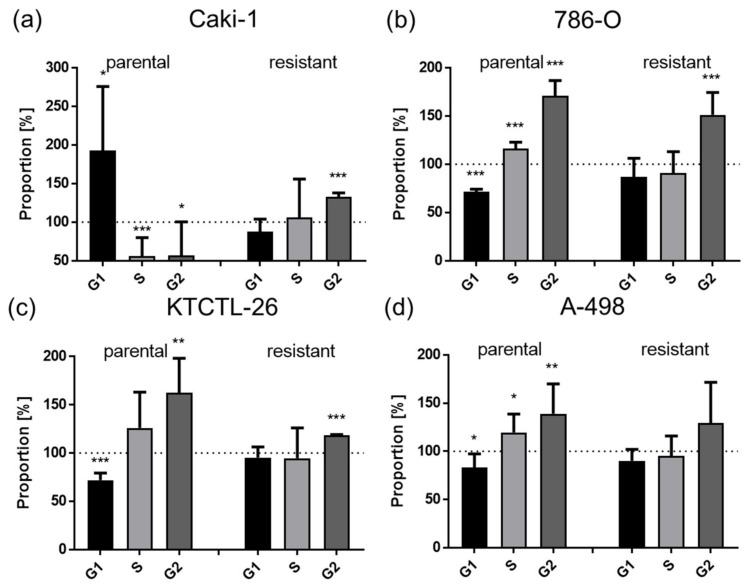
Cell cycle proportionality of parental and sunitinib-resistant Caki-1 (**a**), 786-O (**b**), KTCTL-26 (**c**), and A-498 (**d**) cells in G0/G1, S, and G2/M phases following 48 h exposure to 1.5 µM SHI. Untreated controls set to 100% (dotted line). Error bars indicate standard deviation (SD). Significant difference to untreated control: * *p* ≤ 0.05, ** *p* ≤ 0.01, *** *p* ≤ 0.001. *n* = 5.

**Figure 4 cancers-14-01114-f004:**
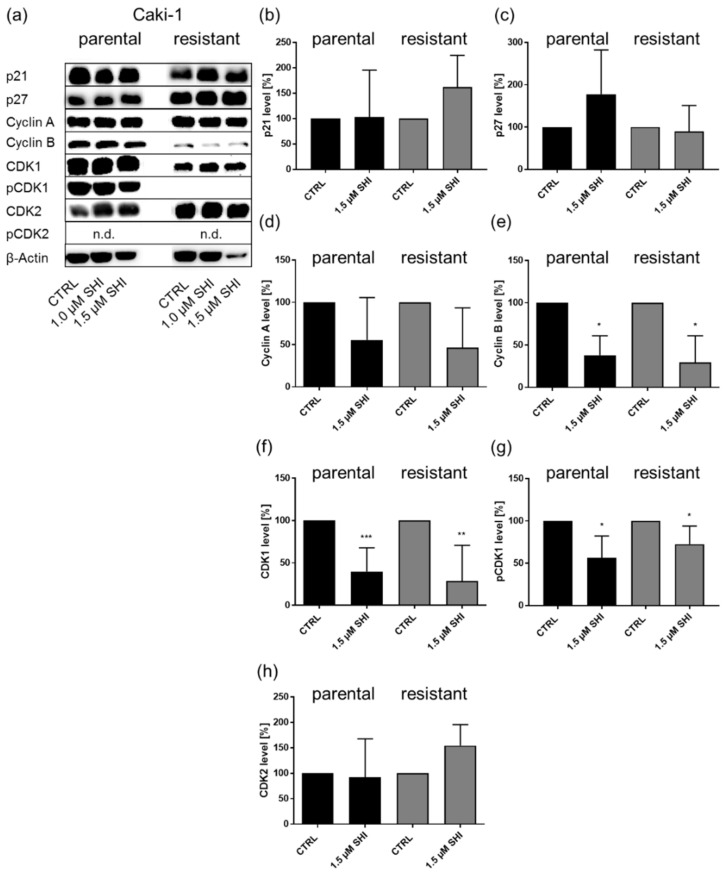
Protein expression and activity of cell cycle regulating proteins in parental and sunitinib-resistant Caki-1 cells after 48 h exposure to 1 or 1.5 µM SHI (**a**). Representative Western blot analysis (*n* = 4). The housekeeping protein, β-actin, served as internal control. Pixel density analysis (Western blot) of the cell cycle regulating proteins p21 (**b**), p27 (**c**), cyclin A (**d**), cyclin B (**e**), CDK1 (**f**), pCDK1 (**g**), and CDK2 (**h**) in parental and sunitinib-resistant Caki-1 cells after 48 h exposure to 1.5 µM SHI, compared to untreated control (set to 100%). Protein analysis was performed with respect to the housekeeping protein β-actin. n.d. = not detectable. Error bars indicate standard deviation (SD). Significant difference to untreated control: * *p* ≤ 0.05, ** *p* ≤ 0.01, *** *p* ≤ 0.001. *n* = 4. For detailed information regarding the Western blots, see Appendix A.

**Figure 5 cancers-14-01114-f005:**
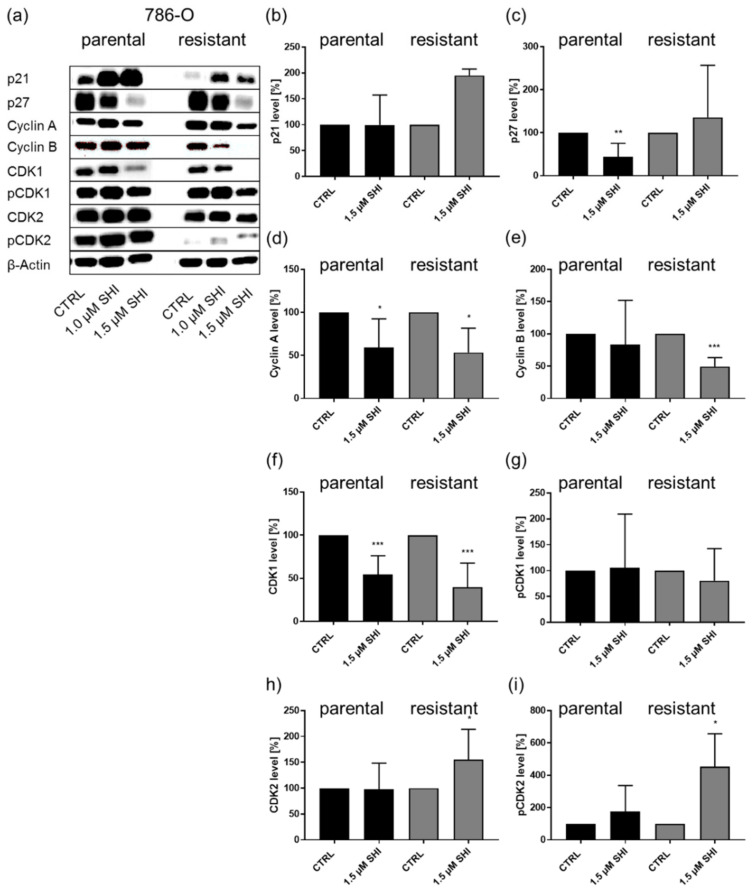
Protein expression and activity of cell cycle regulating proteins in parental and sunitinib-resistant 786-O cells after 48 h exposure to 1 or 1.5 µM SHI (**a**). Representative Western blot analysis (*n* = 4). The housekeeping protein β-actin served as internal control. Pixel density analysis (Western blot) of the cell cycle regulating proteins p21 (**b**), p27 (**c**), cyclin A (**d**), cyclin B (**e**), CDK1 (**f**), pCDK1 (**g**), CDK2 (**h**), and pCDK2 (**i**) in parental and sunitinib-resistant 786-O cells after 48 h exposure to 1.5 µM SHI, compared to untreated control (set to 100%). Each protein analysis was accompanied and normalized by the housekeeping protein β-actin. Error bars indicate standard deviation (SD). Significant difference to untreated control: * *p* ≤ 0.05, ** *p* ≤ 0.01, *** *p* ≤ 0.001. *n* = 4. For detailed information regarding the Western blots, see Appendix A.

**Figure 6 cancers-14-01114-f006:**
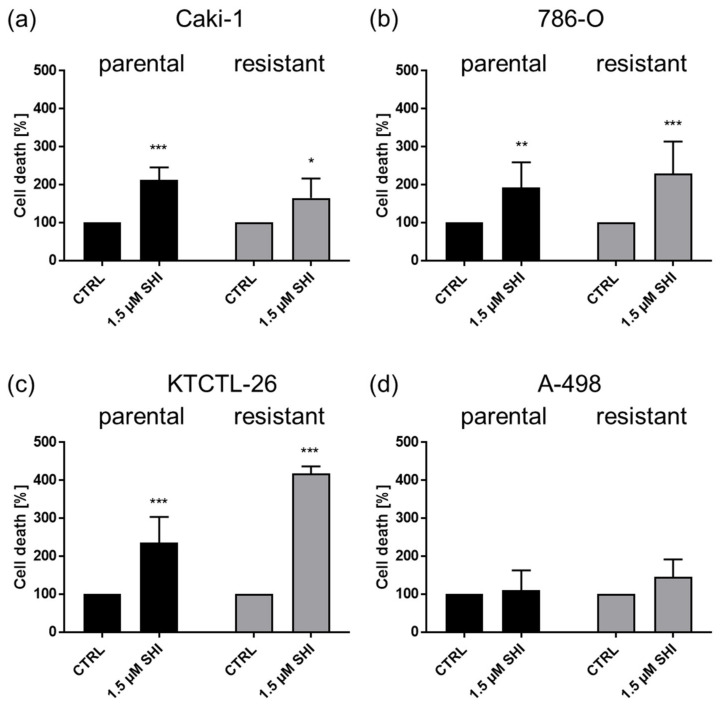
Apoptosis in parental and sunitinib-resistant Caki-1 (**a**), 786-O (**b**), KTCTL-26 (**c**), and A-498 (**d**) cells treated with 1.5 µM SHI for 48 h. Untreated cells served as controls (set to 100%). Error bars indicate standard deviation (SD). Significant difference to untreated control: * *p* ≤ 0.01, ** *p* ≤ 0.01, *** *p* ≤ 0.001. *n* = 3.

**Figure 7 cancers-14-01114-f007:**
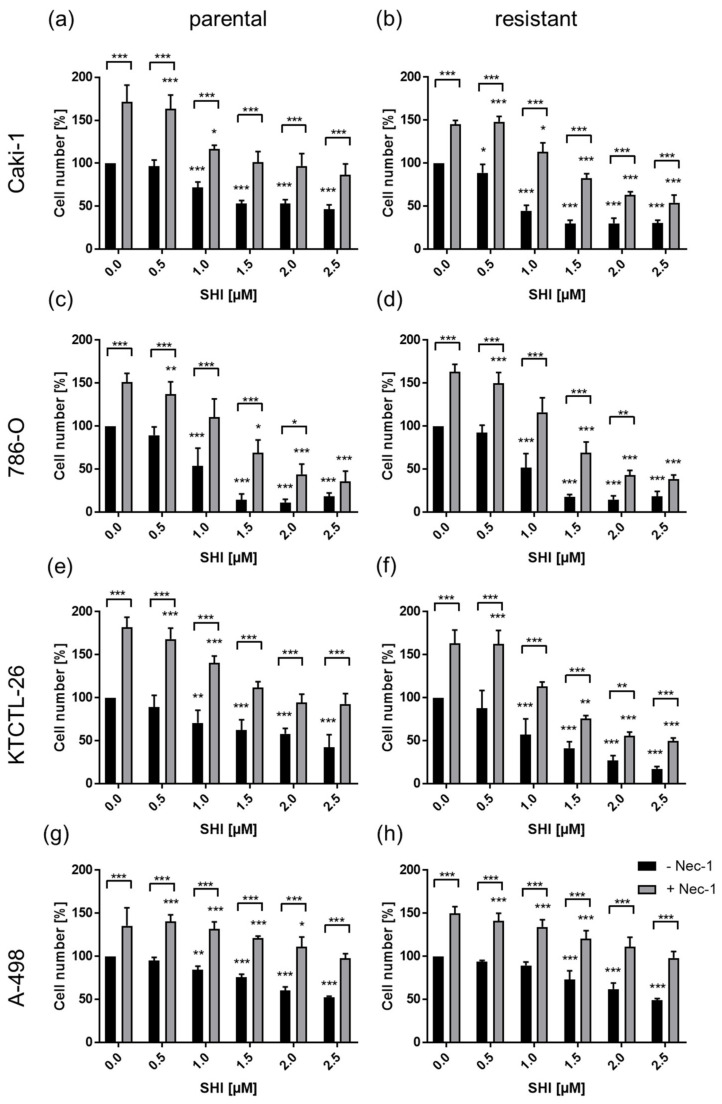
Necroptosis in parental and sunitinib-resistant Caki-1 (**a**,**b**), 786-O (**c**,**d**), KTCTL-26 (**e**,**f**), and A-498 (**g**,**h**) cells treated with 0.5–2.5 µM SHI and 80 µM necrostatin-1 (Nec-1), a necroptosis inhibitor, for 24 h. Error bars indicate standard deviation (SD). Bracketed asterisks indicate a significant difference between Nec-1 untreated and treated cells, unbracketed asterisks indicate significant difference to controls, untreated with SHI (set to 100%): * *p* ≤ 0.05, ** *p* ≤ 0.01, *** *p* ≤ 0.001. *n* = 5.

**Figure 8 cancers-14-01114-f008:**
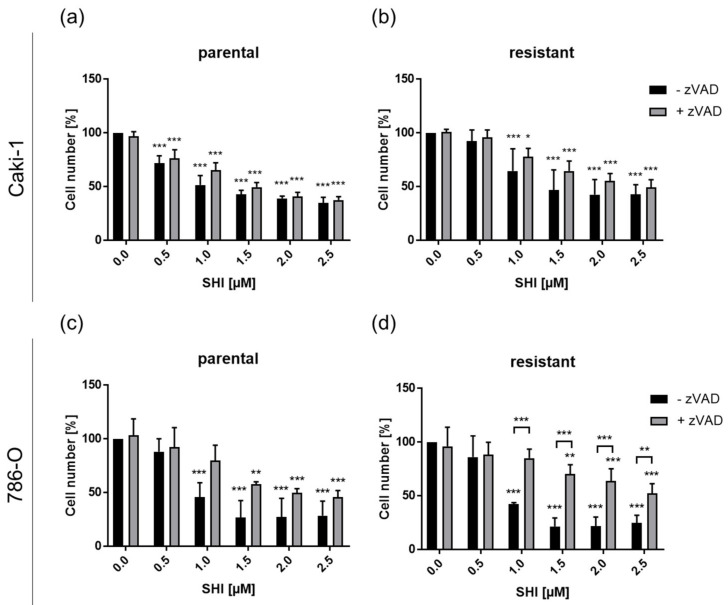
Caspase-dependent or -independent cell death induction in parental and sunitinib-resistant Caki-1 (**a**,**b**) and 786-O (**c**,**d**) cells treated for 24 h with SHI (0.5–2.5 µM) alone or in combination with the multi-caspase inhibitor zVAD (20 µM). Untreated cells served as controls (set to 100%). Error bars indicate standard deviation (SD). Significant difference compared to untreated controls, except for bracketed asterisks, indicating significant difference between untreated and zVAD-treated cells: * *p* ≤ 0.05, ** *p* ≤ 0.01, *** *p* ≤ 0.001. *n* = 3.

**Figure 9 cancers-14-01114-f009:**
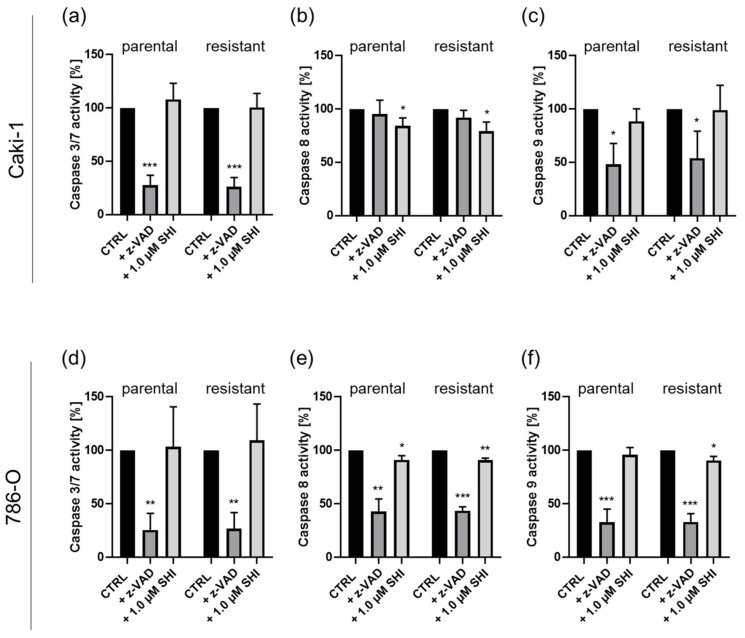
Caspase-independent cell death induction in parental and sunitinib-resistant Caki-1 (**a**–**c**) and 786-O (**d**–**f**) cells after SHI (1.0 µM) or zVAD treatment (20 µM) for 6 h. Caspase activity is expressed as percent, with untreated cells serving as controls (set to 100%). Error bars indicate standard deviation (SD). Significance compared to untreated control: * *p* ≤ 0.05, ** *p* ≤ 0.01, *** *p* ≤ 0.001. *n* = 4.

**Figure 10 cancers-14-01114-f010:**
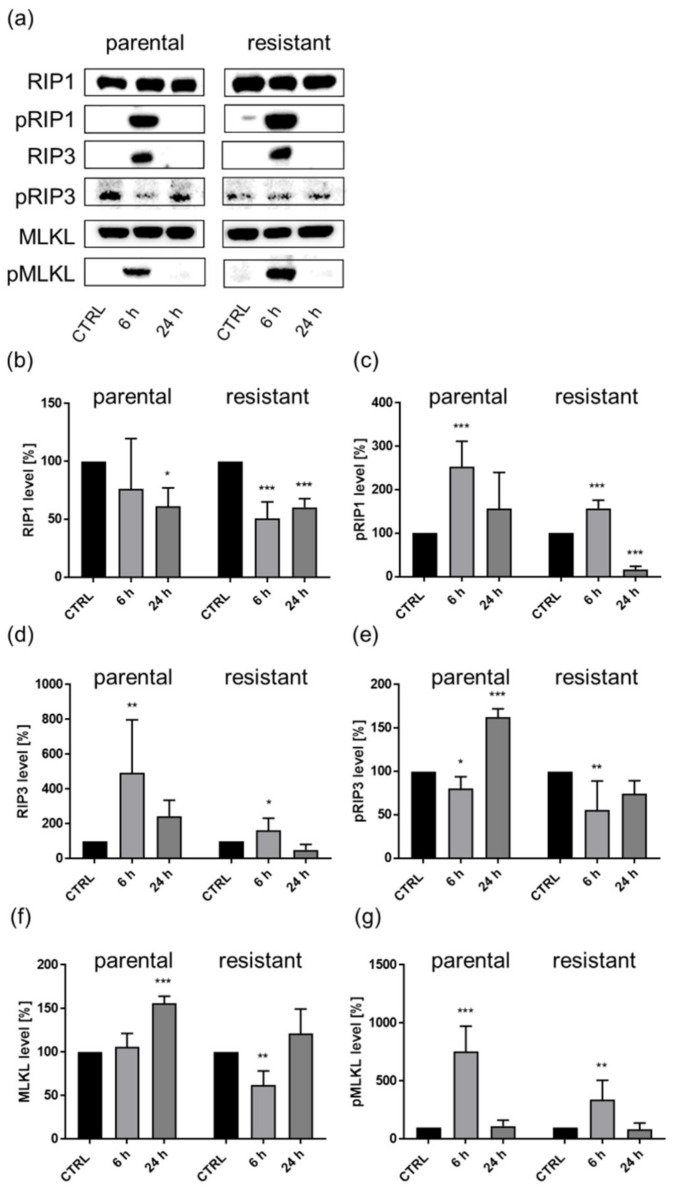
Necrosome complex key proteins in parental and sunitinib-resistant Caki-1 cells after 6 and 24 h exposure to 1 µM SHI (**a**). Representative Western blot (*n* = 3). Pixel density analysis of the necrosome complex proteins RIP1 (**b**), pRIP1 (**c**), RIP3 (**d**), pRIP3 (**e**), MLKL (**f**), and pMLKL (**g**) in parental and sunitinib-resistant cells after 6 and 24 h exposure to 1 µM SHI. Protein analysis was performed by normalization to a total protein control. Untreated cells served as controls (set to 100%). Error bars indicate standard deviation (SD). Significant difference to untreated control: * *p* ≤ 0.05, ** *p* ≤ 0.01, *** *p* ≤ 0.001. *n* = 3. For detailed information regarding the Western blots, see Appendix A.

**Figure 11 cancers-14-01114-f011:**
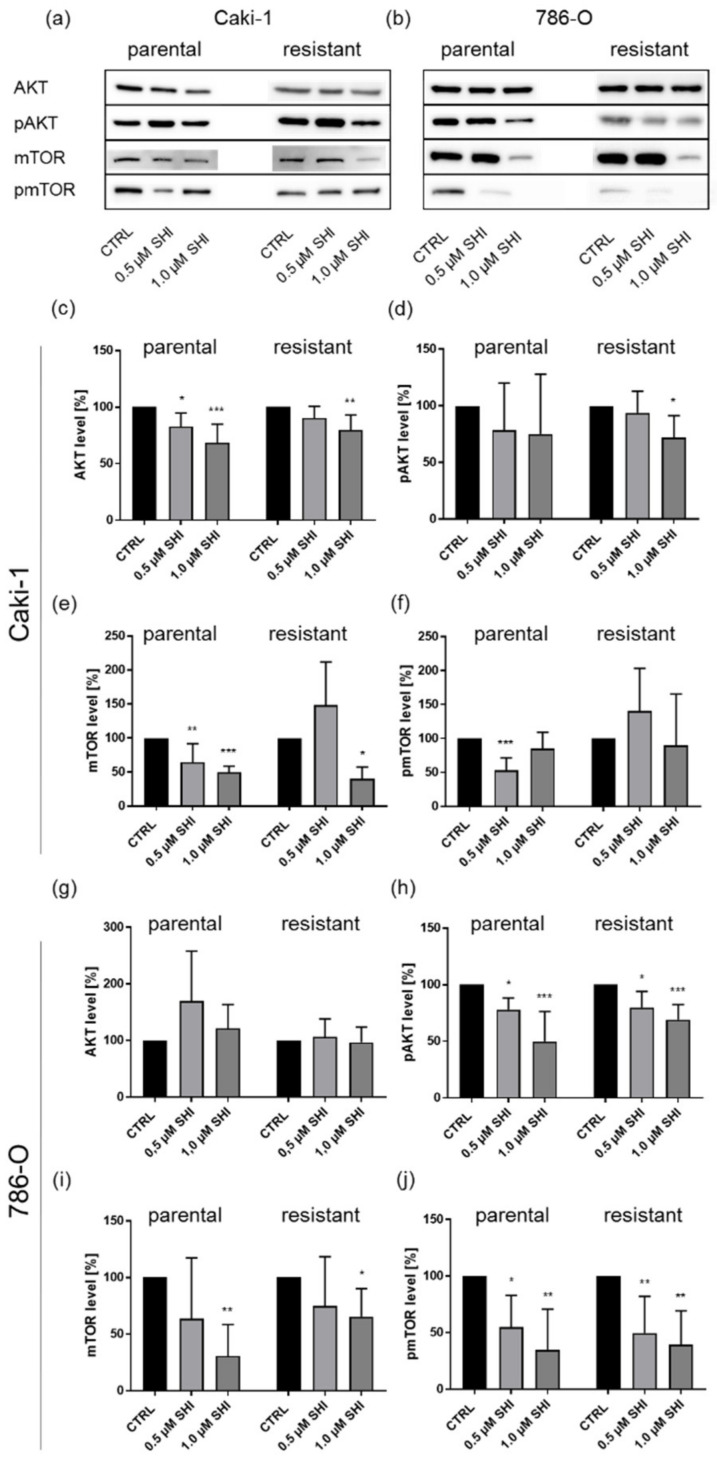
AKT/pAKT and mTOR/pmTOR in parental and sunitinib-resistant Caki-1 (**a**) and 786-O (**b**) cells after 48 h exposure to SHI (0.5, 1.0 µM). Representative Western blot (*n* = 4). Pixel density analysis (Western blot) of AKT (**c**,**g**), pAKT (**d**,**h**), mTOR (**e**,**i**), and pmTOR (**f**,**j**), compared to untreated controls (set to 100%). Pixel density was normalized by a total protein staining. Error bars indicate standard deviation (SD). Significant difference to untreated control: * *p* ≤ 0.05, ** *p* ≤ 0.01, *** *p* ≤ 0.001. *n* = 4. For detailed information regarding the Western blots, see Appendix A.

**Figure 12 cancers-14-01114-f012:**
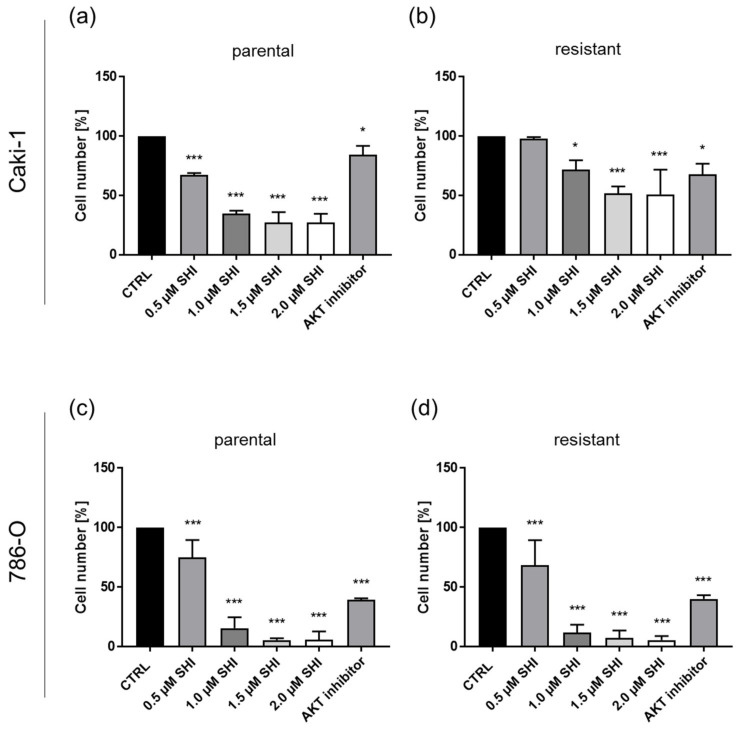
Functional relevance of AKT: growth of parental and sunitinib-resistant Caki-1 (**a**,**b**) and 786-O (**c**,**d**) cells after 48 h incubation with 0.5–2.0 µM SHI or 20 µM AZD5363 (AKT inhibitor). Untreated cells served as control (= 100%). Error bars indicate standard deviation (SD). Significant difference to untreated control: * *p* ≤ 0.05, *** *p* ≤ 0.001. *n* = 3.

**Figure 13 cancers-14-01114-f013:**
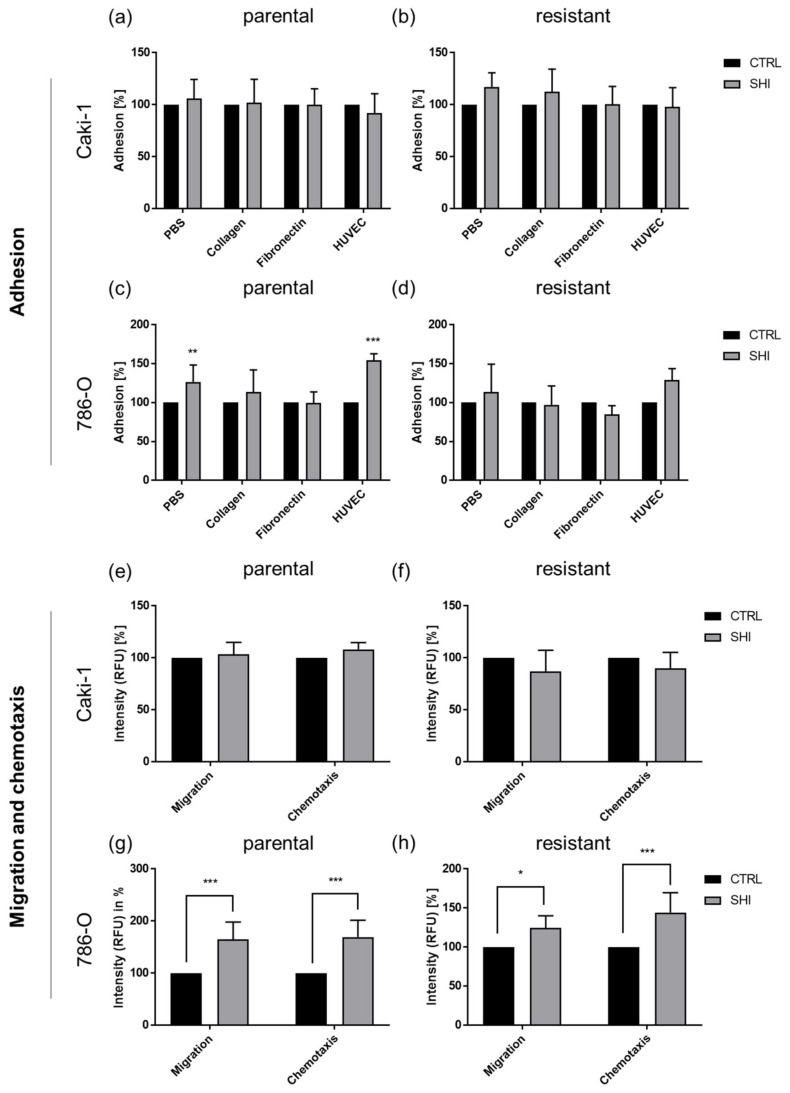
Adhesion capacity and motility of parental and sunitinib-resistant Caki-1 (**a**,**b**) and 786-O (**c**,**d**) cells after 72 h exposure to 1 µM SHI. Adhesion to extracellular matrix (ECM) proteins, collagen, and fibronectin as well as to vascular endothelium (HUVEC) (**a**–**d**), compared to unspecific adhesion (PBS) to plastic (set to 100%). Migration and chemotaxis of parental and sunitinib-resistant Caki-1 (**e**,**f**) and 786-O cells (**g**,**h**). Untreated cells served as control (set to 100%). RFU = relative fluorescence units. Error bars indicate standard deviation (SD). Significance compared to untreated control: * *p* ≤ 0.05, ** *p* ≤ 0.01, *** *p* ≤ 0.001. *n* = 5.

## Data Availability

All data generated for this study are included in the article or Appendix A.

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
