# Peer review of "Shikonin Inhibits Cell Growth of Sunitinib-Resistant Renal Cell Carcinoma by Activating the Necrosome Complex and Inhibiting the AKT/mTOR Signaling Pathway"

_cancers, 2022, doi:10.3390/cancers14051114_

Round 1
Reviewer 1 Report
In this manuscript, Markowitsch et al., investigate the effects of shikonin on parental and sunitinib-resistant renal cell carcinoma (RCC) cells. They found that shikonin suppressed the growth, proliferation capacity and clonogenicity of both tumor cell populations. It also induced necroptosis and downregulated the Akt/mTOR pathway. Shikonin has been shown to elicit similar effects on other tumor cells. Thus, this study is not very novel. Nevertheless, these data are of interest to the scientific community, and I will recommend the publication of this manuscript, provided that the authors address the following comments.
Major comments:
- The authors need to explain how they induced and cultured sunitinib-resistant RCC cells.
- Why did the authors use untreated cells as controls? Shikonin is soluble in organic solvents. Is it possible that the solvent might influence tumor cell behavior?
- In figures 4,5,10, the authors performed image analysis to quantify protein intensity. However, the signal of most of the bands was saturated and saturated bands hide actual variation in protein levels and underestimate the amount of protein present. Moreover, in figures 10 and 11, the authors must use a proper loading control. Also, in figure 11, the authors claim that shikonin downregulated the activity of AKT and mTOR. To prove this, they need to normalize pAKT and pmTOR to AKT and mTOR, respectively.
- Why does shikonin evoke G0/G1 phase in the parental Caki-1 cells?
Minor comments:
- The line plots in Figure 1 are very difficult to distinguish.
- In materials and methods, the authors state that they studied cell motility through 0.8 μm pore size. Do they mean 8 μm pore size?
Reviewer 2 Report
The authors presented that shikonin, from traditional Chinese medicine, had an inhibitory effect on renal cell cancer proliferation through cell cycle arrest and necroptosis. The mechanism of this molecule was investigated in detail in vitro on four different cell lines, each cell line was sensitive or resistant to standard chemotherapy Sunitinib, and they showed the efficacy and safety of Shikonin. Especially in therapy resistant renal cell cancer cell lines, shikonin reduced renal cell cancer cellular proliferation in a low µm-range and inhibited colony formation. The authors have done many experiments to prove facts above. Shikonin seems to be useful for a therapeutic option against HCC, but I have several concerns and questions about this manuscript.
Major
- In Figure 1, the authors presented dose and time dependent inhibition of proliferation of several cell lines. IC 50 Results of each cell line were shown. In Figure 1i the IC50 value in A498 cells is given as 1.54 µM. But in the graph the line is always above 50 % cell number in the highest concentration used. How does the authors calculate these IC50 values?
- It would be helpful for the reader to compare your data with sunitinib as a standard chemotherapy.
- In Figure 4 and 5, ß-Actin was used as housekeeping protein. But it seems that this protein is already regulated by Shikonin. Please use another housekeeping protein, to avoid unnecessary calculation bias as seen in p21 expression in 786-O cells. Here the western blot shows upregulation of p21 but the quantification shows no change. Also p27 expression in the resistant cell line seems to be downregulated, but the quantification shows no change.
- The authors explained that necroptosis is a mechanism of cell death and that “Treatment with necrostatin-1, a necroptosis inhibitor, significantly abolished the growth inhibitory effect of SHI in parental and sunitinib-resistant… cells”. But how the authors explain, that after necrostatin treatment, also the non-treated control cells seems to proliferate more? I would assume that normalization of control cells would help to figure out this calculation bias.
- In figure 10 the house keeping protein in missing. Please provide a normalization protein on the western blots.
- There is also a study of Tsai et al DOI: 10.3390/antiox10111831 published last year showing that “Shikonin Induced Program Cell Death through Generation of Reactive Oxygen Species in Renal Cancer Cells”. Please use this literature in the discussion and compare.
Reviewer 3 Report
The manuscript titled “Shikonin Inhibits Cell Growth of Sunitinib-Resistant Renal Cell Carcinoma by Activating the Necrosome Complex and Inhibiting the Akt/mTOR Signaling Pathway” is comprehensive and well prepared. The authors investigated a traditional Chinese medicine shikonin(SHI) that has anti-tumor properties. In the current study, the authors tested the efficiency of SHI on the therapy-sensitive and therapy-resistant advanced renal cell carcinoma (RCC) cell lines. Sunitinib-sensitive and sunitinib-resistant Caki-1, 786-O, KTCTL-26, and A-498 RCC cell lines were used to extend their previous discovery.
However, there are still some concerns and the comments have been pointed out as following that the authors may want to consider.
Comments:
1: For all the figure legends, please consider making them consistent.
“* p ≤ 0.05, ** p ≤ 0.01, *** p ≤ 0.001” is in the figure legends of Figure 1, Figure 2, Figure 3, Figure 4, Figure 5, Figure 6, Figure 7, Figure 8, Figure 9, and Figure 11.
“* = p ≤ 0.05, ** = p≤ 0.01, *** = p ≤ 0.001” is in the figure legends of Figure 10 and Figure 13.
“* = p ≤ 0.05, ** = p < 0.01, *** p < 0.001” is in the figure legend of Figure 12.
2: Line 97: “Human umbilical vein endothelial cells (HUVEC) cells” one of the “cells” should be deleted.
3: Line 149: “1 × 104 cells/well were seeded onto a 96-well-plate”. The cells were growing directly in the well of the plate. It should be into instead of onto.
4: Line 156: “~1 h at 120 V”, does “~” indicates “around 1 h”?
5: Line 203: “1.25 x 105 HUVEC’s were plated onto 24-well-plates 16 h prior to an adhesion assay.” It should be into instead of onto. And I’d suggest authors remove ’ from HUVEC’s.
6: Figure 10a: Would you please explain why the RIP3 was not detected in the control groups, while the pRIP3 was detected? Or the images were accidentally switched between pRIP3 and RIP3.
Round 2
Reviewer 1 Report
The authors partially addressed the reviewer's concerns. However, I am still concerned about the quality of the western blots and the use of untreated cells as controls. The issues below need to be addressed prior to publication. Specifically:
1) They did not provide any evidence that DMSO doesn't influence tumor cell behavior. The authors state: "dilution of DMSO was at least 1:4,000. Thus, the impact of DMSO is negligible and we therefore focused on the untreated cells as controls." This is not a proof. A side-by-side comparison needs to be performed in order to verify that DMSO-treated cells behave the same way as untreated cells.
2) The authors need to indicate which of the supplementary blots were used in the manuscript. It seems that the authors have cut and paste bands to bring them closer to other bands (e.g., p21, p27, CAKI-1, CDK2). Why didn't the authors label all lanes? The authors need to run new blots with samples placed next to each other. Also, they need to better align the labels with the bands.
Reviewer 2 Report
I have no further comments to the authors.
Author Response
Our answer: Thank you very much.

Round 3
Reviewer 1 Report
The authors have addressed my concerns